# Public Mixed Funding for Residential Aged Care Facilities Residents’ Needs in the Asia–Pacific Region: A Scoping Review

**DOI:** 10.3390/ijerph20217007

**Published:** 2023-11-01

**Authors:** David Lim, Ashley Grady, Karen P. Y. Liu

**Affiliations:** 1Centre for Improving Palliative, Aged and Chronic Care through Clinical Research and Translation, University of Technology Sydney, Ultimo, NSW 2007, Australia; 2School of Health Sciences, Western Sydney University, Campbelltown, NSW 2560, Australia; 3Department of Rehabilitation Sciences, The Hong Kong Polytechnic University, Hung Hom, Hong Kong; karen.liu@polyu.edu.hk

**Keywords:** public funding, care needs, residential aged care, Asia–Pacific, aging

## Abstract

Due to population aging and sociodemographic change, there is an increasing reliance on residential aged care facilities in the Asia–Pacific region. Most countries have adopted taxation as the primary means to levy capital for funding universal health services and means-testing of benefits may be further incorporated as a policy balance between horizontal equity and fiscal sustainability. It was hypothesized that residential care needs are evaluated by assessments relating to funding; this scoping review seeks to synthesize how such assessments relate to the care needs of residents. Searches were conducted in concordance with a priori protocol for English-language literature published since 2008 in Embase, CINAHL, PubMed, Scopus, JBI, TROVE, and four peak international organizations for studies and reports that describe the assessment of residents’ needs in Asia–Pacific countries that used a mixture of taxation and means-testing to publicly fund residential aged care. One paper and 47 reports were included. Australia, New Zealand, and Singapore utilize a taxation and means-tested user charge approach to fund residential aged care needs. The common care needs assessed include health conditions, daily activities, cognition, psychiatric, and behavioral needs. While essential care needs are publicly funded, other holistic care needs, such as spirituality and autonomy-based needs, still need to be covered for meaningful occupation by the residents.

## 1. Introduction

Universal health coverage allows people to access the health services they require across a range of promotive, preventative, curative, rehabilitative, and palliative amenities without encountering financial strain to pay for their care [1]. While there are innumerable methods by which universal health care has been established across different countries, there are common traits that are required to achieve this feat, including the creation of public programs to extend healthcare access, improved equity, and the pooling of financial risk from government subsidies, taxes, and/or mandatory insurance schemes to reduce out-of-pocket payments [2]. The cost of aging in relation to healthcare expenditures is often a concern for governments [3]. This concern is increasing globally as the number of people aged 60 years or older is projected to rise to 2 billion, or 22%, of the total population by 2050, compared to 900 million, or 12%, in 2015 [4]. Moreover, the costs of delivering elderly care services are also rising [5]. While many countries hypothecate, fully or partially, aged and long-term care through the public sector [6], one method that has been employed to efficiently manage health financing for aged or long-term care and ensure horizontal equity and fiscal sustainability is means-testing. This is a process by which income and/or asset tests are conducted to determine eligibility thresholds for the publicly funded government subsidy or determine if the individual is required to pay out-of-pocket for their care [5]. The individuals and/or their families/estates thus pay a small portion of the care at the point of use. This means-tested approach may be used to determine user charges in conjunction with taxation to fund Residential Aged Care Facilities (RACFs). This approach is multifaceted. Taxation involves governments or public insurers garnering revenue through general income tax or levies to fund public expenditure [7]. This taxation is then supplemented by means-tested user charges, which comprise mandatory or voluntary levies paid by people who benefit from the publicly funded RACFs [8]. Countries that utilized such mixed or progressive funding for RACFs include France, Austria, and some regions in Italy [6].

The funding model of the aged care sector within the Asia–Pacific (APAC) region is of particular interest due to population aging and sociodemographic change [9]. It is estimated that over 1.3 billion older people will be in this region by 2050 [10]. On a global scale, this rapid aging and its subsequent management need to be considered, as the APAC region constitutes 3.45 billion, or 53%, of the world’s total population across 48 countries within Southeast Asia, the Western Pacific, and Oceania [11]. By 2050, it is projected that 11 countries in the APAC region will be qualified as a “super-aged society” (versus only Japan in 2020), with the most aged populations being in Japan, the Republic of Korea, and Singapore [12]. Much of the APAC region has traditionally relied on informal, family-based care due to cultural values, filial piety, and the stigma associated with placing elderly family members in RACFs [13]. This unpaid, family-based, informal care is typically provided by the wives or daughters of the elderly person or by hired domestic helpers within the home; it often involves the completion of domestic duties and assistance with personal care [14,15]. However, rapid population aging in the APAC region, in combination with sociodemographic change, the increased incidence and prevalence of chronic conditions and multiple co-morbidities, and the complexity of caring for complex chronic conditions, has led to an increased need and reliance on RACFs. In 2018, an estimated 1,404,146 people were residing in RACFs across 13 APAC countries that are members of the Organization for Economic Co-operation and Development; this number is projected to significantly increase in the coming decades [16].

The focus on means-testing in this review stems from the basic principle of social policy: the just allocation of scarce resources based on determinants of health and the demands of responsible governments to balance competing fiscal demands. Thus, by describing the current funding for residential aged care facility residents’ needs in Asia–Pacific countries that provide public mixed funding, this review is forward-looking and aims to inform further discourse on future funding models for RACFs in the APAC region.

## 2. Methods

This review was conducted under the a priori protocol [9]. The databases searched included Embase (OVID), CINAHL (EBSCOhost), PubMed (NLM), Scopus (Elsevier), and JBI (OVID). The gray literature was sought through TROVE and relevant health organizations and government departments, including the World Health Organization, United Nations, Asian Development Bank, and Organization for Economic Co-operation and Development. A hand search of the reference lists of the included papers was also conducted to find additional potentially relevant papers. The papers included have been published in English since 2008. The Population/Concept/Context logic grid was used for the search. Due to the varied eligibility criteria for citizens to access RACFs and receive public, means-tested assistance to fund their care in these facilities across the APAC countries, a specified age range was not applied. RACFs were defined broadly as any institutional facility that adults reside in while receiving care that sustains health or assists in completing Activities of Daily Living (ADLs) to meet residents’ care needs. Care needs were defined as any aspect of a person’s functioning, including physical, cognitive, and behavioral traits requiring intervention or assistance, necessary to sustain health, complete ADLs, or for meaningful engagement.

The initial searches were conducted in July 2021, and an updated search was conducted in September 2022. Two review authors (A.G. and D.L.) assessed the identified papers for eligibility against the selection criteria by title, abstract, and then full text. There was a high level of agreement (>90%) between reviewers. Any disagreement was resolved through discussion and with the involvement of a third reviewer (K.L.). Two researchers (A.G. and D.L.) extracted data from the included papers using an adapted version of the JBI Manual for Evidence Synthesis data extraction instrument [17]. Any disagreement was resolved through discussion and consultation with the third reviewer (K.L.). To supplement the data extracted, all corresponding authors, relevant health organizations, and government departments were contacted via email (A.G.) with a request to direct the reviewers to further relevant information. Two researchers (A.G. and D.L.) completed quality appraisals for the included papers using the JBI critical appraisal tools descriptively. A complete agreement was reached for the appraised papers.

## 3. Results

### 3.1. Search Results

A total of 2341 potential papers and reports were retrieved; 2157 were sourced from the databases and 141 from relevant health organizations. After the removal of duplicates, 2114 records from databases and 141 reports were screened for eligibility. One paper and 47 reports were eventually included in this review. No additional papers were retrieved from hand searches of reference lists. The Preferred Reporting Items for Systematic Reviews and Meta-analyses extension for scoping reviews (PRISMA-ScR) flow diagram [18] was used to present the search results and the study inclusion process in Figure 1. 

### 3.2. APAC Countries That Utilize Both Public Taxation and Means-Testing User Charges to Fund RACFs

It was noted that Australia, China, New Zealand, Japan, and Singapore offer publicly funded RACFs that require residents to pay user charges [9]. However, in Japan, access to publicly funded RACFs is not means-tested, with the user charge set to 10% regardless of income [19]. Across China, the Chinese Central Government requires provincial governments to provide sufficient funds for long-term elderly care, resulting in different funding models used across the country [20]. While the Jiangsu province utilizes a means-tested model [21], there is insufficient information to answer the subsequent questions of this review. Therefore, Australia, New Zealand, and Singapore were included in the subsequent analysis.

### 3.3. Assessments Related to Funding

#### 3.3.1. Australia

Australia utilized the Aged Care Funding Instrument (ACFI) between 2008 and 2022 to adjudicate funding based on a resident’s care needs [22,23]. Those with high needs would receive a greater subsidy. It was a means-tested user charge assessment to determine if the resident was required to pay an additional charge to contribute to their care. The ACFI process contained five steps: the first being assessment, checklist (step 2), rating A–D, in which the assessed care needs were assigned a rating that dictated the amount payable per resident based on the amount of assistance required to complete the task in question (step 3), submission to the Department of Human Services as an ACFI claim (step 4), and the fifth stage related to ensuring that records were kept in the appropriate manner and for the required time (three years) so that the Department of Human Services can review information if needed. ACFI examined care needs across three domains: ADL, behavior, and complex healthcare. These domains were subdivided into 12 areas of care needs: nutrition, mobility, personal hygiene, continence, cognitive skills, wandering, verbal behavior, physical behavior, depression, medication, and complex healthcare procedures. Additional assessments, such as the Psychogeriatric Assessment Scales—Cognitive Impairment Scale and the Cornell Scale for Depression, might be used. While assessors were trained to complete the ACFI, the accuracy of the results may potentially be limited as additional assessments used to determine ratings for some questions required additional training to complete properly, and it was unclear if this training was provided to all aged care staff, thus there are potential impacts on the inter-rater validity and accuracy of the results. Additionally, the ACFI was not designed to inform care planning but rather was designed to assess the degree of care each aged care resident needs to allocate funding accordingly, thus there is low clinical utility.

There are four levels of funding for each domain: nil, low, medium, and high. The subsidy RACFs received for each individual relate to the individual’s combined payable amount for each domain. The means-testing occurs through the Department of Human Services; it was used to determine if the resident is required to pay an additional means-tested user charge to contribute to their care [24]. This involved an assessment of income, including support payments from the Australian Government such as aged pension or income support supplement, income from gifting or financial investments, assets including household contents and personal effects, investment properties, and superannuation [25]. The user charge and the ACFI-based subsidy are used to calculate the residential care costs in Australia and comprise 85% of the revenue received by residential care facilities [26].

During this review, the new AN–ACC Assessment Tool replaces the ACFI as of 1 October 2022. The AN–ACC assessment tool focuses on physical ability (mobility), cognition, behavior, and mental health [27]. The AN–ACC classification informs 13 classes of care funding, an expansion from the previous four categories. The assessment is conducted independently by Assessment Management Organizations. The AN–ACC assessment tool incorporates: a 4-item scale; Resource Utilization Groups—Activities of Daily Living that measures motor functions with ADLs for bed mobility, toileting, transfer, and eating; the Australian-modified Karnofsky Performance Scale which measures overall ability to perform ADLs; Rockwood frailty scale and frailty (falls and weight loss) checklist; the Braden scale for predicting pressure sore risk; the modified De Morton mobility index; an Australian Functional Measure for care burden, based on the Functional Independence Measure; and the Behavior Resource Utilization Assessment to document the implications of residents’ behavior for caregivers and service providers. RACFs can seek reclassification of assessment decisions. 

The AN–ACC subsidy comprises a fixed base care tariff (based on rurality and indigenous status) and variable funding based on the AN–ACC classification. There are 13 funding categories, ranging from palliative care (class 1: AUD 216.80 per day), independent with no compounding factors (class 2: AUD 41.19 per day), not mobile, low functions, and high-pressure sore risk with compounding risks (class 13: AUD 216.80 per day) [28]. The new AN–ACC does not change the threshold means-testing requirement.

#### 3.3.2. New Zealand

The needs assessment determines the level of need the person requires; this is rated on a 5-point scale from very low to very high [29]. The needs assessment conducted will be the interRAI Contact Assessment if the person is considered to have non-complex needs, as signified by the referral or assessment that occurs through the interRAI Home Care Assessment when the person appears to have or is known to have complex needs. These needs assessments are conducted to ensure that the person’s care needs are known and to confirm that community options to meet the person’s needs are not available or suitable. People assessed as having high or very high needs are eligible to enter residential care. The care needs of RACF residents are evaluated by the International Resident Assessment Instrument for Long-Term Care Facilities (interRAI LTCF) assessment [30]. This assessment is related to care needs but does not align with funding. This assessment addresses care needs, including but not limited to cognition, communication and vision, mood and behavior, psychosocial well-being, functional status, disease diagnosis/conditions, health conditions, and oral and nutritional status. The assessment is performed by a registered nurse; it requires direct questioning of the resident and their primary support person, as well as observation of the resident and review of their medical records. The interRAI LTCF is said to provide reliable data, with results indicating that 93.8% of items assessed achieved substantial reliability [31]. There may be potential variations as to how the assessor is specifically funded to undertake the assessment.

Once a person has been assessed as needing long-term residential care, they are given an application for the Residential Care Subsidy. The applicant needs to undergo the financial means-test. The first component (asset test) assesses the value of the person’s non-exempt assets to determine if their assets exceed, meet, or are below the asset threshold [32]. People who exceed the asset threshold must pay the maximum contribution, while people who are determined to meet or be below the asset threshold pay a means-tested user charge based on the second component (income assessment test). The income assessment considers the income of the person or the person’s spouse or partner, any benefit received by the person, and 50% of any pension received by the person or their spouse/partner. The financial means assessment is only concerned with determining if the person applying for a subsidy can financially contribute to the cost of their long-term residential care; consequently, the subsidy provided is based on the results of the means-test, not on the resident’s medical or care needs.

#### 3.3.3. Singapore

To be eligible in Singapore for subsidized nursing home placement, the prospective resident must have trialed alternate care arrangements, including employing domestic assistance, attending daycare, or receiving home care, which was unsuccessful and indicated that the person could not have their care needs met at home [33]. The person must have a physical or mental disability and reduced mobility, in which case they may be semi-mobile, use a wheelchair, be bed-bound, and require assistance with ADLs. People eligible to enter residential care provided in Voluntary Welfare Organizations Nursing Homes and require subsidy from the Ministry of Health to assist in funding their care must pass the household means-test [34,35]. This means-test considers the total gross monthly income of the resident and any family members over the age of 21 living in the household, the number of family members the resident has, and any major assets that the resident owns. Another factor that impacts a resident’s subsidy is the results of their needs assessment [36]. The Resident Assessment Form (RAF) is used: category 1 patients are physically and mentally independent; those who are semi-independent are classified as category 2; people who are wheelchair or bedbound are classified as category 3; and highly dependent people are classified as category 4. Category 1 and 2 patients are generally admitted to assisted living facilities, while category 3 and 4 patients are eligible for placement in nursing homes [37]. The RAF assesses the person’s functional status and signifies the resident’s current clinical care requirements to classify the resident by their care needs and how much nursing time will be required [38]. There are nine care need indicators: mobility, feeding, toileting, personal grooming and hygiene, treatment, social and emotional needs, confusion, psychiatric problems, and behavioral problems. This RAF classification influences the funding levels they receive; the highest level of subsidy is provided to people with the highest care needs (category 4). This assessment is conducted by a nurse, a nurse case manager, or a doctor [36].

#### 3.3.4. Summary

Australia and Singapore use two assessments to determine funding levels: a means-test and an assessment that bases funding on care needs; New Zealand uses one assessment to determine funding based on the resident’s financial circumstances.

The ACFI/AN-ACC (Australia), the InterRAI LTCF (New Zealand), and the RAF (Singapore) used in the three countries categorize the care needs differently. The common care needs identifiable across the three assessments include health conditions, performance in ADLs (mobility, feeding/nutrition, toileting, personal hygiene, and grooming), cognition, psychiatric needs (including wandering/absconding), and behavioral needs. Other care needs identified by individual assessments are nutrition, oral status, medication, and complex health care procedures. In the case of Australia and Singapore, these assessments are also used to determine funding levels. In New Zealand, the subsidy is based on means-testing only.

## 4. Discussion

The APAC region is at the forefront of population aging globally [10]. While the region is diverse, a common thread across many of the APAC countries is the strong reliance on traditional, informal, family-provided care based on the prevailing cultural values of family-based care; for example, in China [39], Mongolia [40], Bangladesh [41], Pakistan [42], Indonesia [19], Sri Lanka [43], and Fiji [44]. Governments appeal to filial cultural values and proactively encourage families to provide support for their elders and family-based care [15]. In some APAC countries, including China [20] and India [45], aged care being the responsibility of the older person’s family is stipulated within relevant policies and legislation. The region is also experiencing sociodemographic changes—for example, increased female education, advancement, and workforce participation; greater movement for employment and living standards; a transition from consanguineal to conjugal family structure—impacting the provision of gerontological care and increasing reliance on RACFs for routine and complex care.

In the absence of a universally accepted definition of RACFs and what RACF residents’ care needs should be [46], this lack of a globally accepted definition increases the difficulty of understanding and meeting these RACF residents ‘care needs [21,47]. This in turn has direct implications for how RACF care needs are assessed and funded. The United Nations Sustainable Development Goal 3 of “wellbeing for all, at all ages” imposes on its member states to strive towards universal health coverage for older adults [48]. The World Health Organization’s Global Strategy and Action Plan on Aging and Health identified that older adults have less capacity to pay for RACF services and called for the strengthening of policy and transformative reform to address the needs of older adults [48]. Some countries have incorporated the means-testing of private health in the allocation of public care as an attempt to strike a balance between equity and sustainability. The notion of “user pay” is congruent with the neoliberal government approach and perception of fairness [49]. It was further proposed that if “citizens believe that their taxes are being used properly, they are more likely to support higher taxes” [50], (p. 39). The latter may be of additional relevance to the APAC region, with 17 out of the 21 member states considered to be least developed or developing [51], as the region prepares itself for population aging and coupling with the current cost of living crisis.

In Australia, the recent Australian Royal Commission into Aged Care Quality and Safety identified issues surrounding means-testing and equity in the public funding of RACFs [52]. This Royal Commission was tasked by the Australian Government in late 2018 to investigate the quality, funding, and needs of aged care services after a series of high-profile media reports of substandard care and abuse and systemic failure in the aged care sector [53]. The Commission, which reviewed over 10,500 public submissions and heard from 641 witnesses, concluded that the current system disproportionately impacts people with medium-level resources compared with people who have higher levels of private wealth. The identified issues surrounding the equity of the means-testing system used in Australia to fund RACFs and the costs of care within such facilities justified investigating current and future models of public funding for RACFs. The current and replaced assessments used to determine a resident’s subsidy are based on a medicalized model and closely linked to the assessments for care needs. The AN–ACC assessment has expanded on the ACFI; the costs of subsidy are weighted to the National Weighted Activity Unit to take account of variations in the costs of providing care, splitting the subsidy into a fixed component (facility-based care) and a variable component based on the individual residents’ AN–ACC assessment of care needs. Means-testing remains the threshold to ascertain a resident’s eligibility.

Likewise, in Singapore, two assessments influence the subsidies provided to RACFs: the means test and the RAF. It was also discovered during this review that the RAF was developed based on the ACFI’s predecessor, the Residential Classification Scale [54]. The Residential Classification Scale was used in Australia from 1997 to 2008 as a resource allocation instrument for RACFs. It determines the degree of dependence a resident displays regarding physical and social functioning and the level of supervision or assistance required [55]. This may be why there are similarities between the assessments used in Australia and Singapore.

The findings of this scoping review do affirm that RACF care needs identified by the assessments used in Australia, New Zealand, and Singapore align with the care needs recognized as important by RACF residents within the literature [47,56]. While essential care needs are indeed being addressed, an improvement could still be made by adding additional or greater weight to components of the assessments that are congruent with meaningful occupations such as spirituality, recreational needs, self-esteem, and autonomy-based needs to ensure that the needs are assessed more holistically and not solely based on the biomedical model of health or the deficit model.

A qualitative study found that nursing home residents identify their care needs according to six themes, referred to as BEEMPS: the Body, Economics, Environment, Mind, Preparation for death, and Social support [47]. Residents in this study identified their care needs to include their bodily limitations and the need for assistance with ADLs and nursing care; their economic conditions relating to paying fees for the care received; the cleanliness of the environment and availability of space; their psychological health and emotional needs; their need to make arrangements before they pass away; and their need for social support and the ability to participate in recreational activities. Therefore, the care needs of RACF residents can be defined as any aspect of a person’s functioning, including their physical, cognitive, and behavioral traits, that are necessary to sustain health or complete ADLs. Residents prioritize their care needs to include the need for assistance to maintain participation in daily activities, timely access to assistance, the management of medical conditions, the involvement of family in care, maintaining a sense of spirituality, and being treated with respect by staff members [56]. These elements are holistic, demonstrating that residents consider their care needs to be associated with physical health as well as their spiritual, psychological, emotional, psychosocial health, and well-being; the suggestion that RACFs must ensure that care needs are assessed holistically, especially residential care, is more than the provision of health and well-being care services, and RACFs is more than a place where care services are delivered. If RACFs are to truly become the older adults’ home and substitute for family life, both health and social service lenses are required in considering the types of RACF care services required for a resident to lead a meaningful and engaging life. Furthermore, understanding and fulfilling these spiritual, psychosocial, emotional, cultural, social, as well as physical care needs may improve residents’ outcomes and foster the development of rapport between staff and residents, allowing residents to self-advocate for their care through the discussion of their care needs formally through assessment. This may reduce the vulnerability and power imbalance experienced by some RACF residents.

The APAC region consists of diverse cultural, political, economic, and religious landscapes. Globalization has seen greater migration and regional cooperation between countries. Lessons learned from and about APAC countries in funding and delivering RACF care, especially in the current post-COVID-19 pandemic and cost of living crisis era, may assist in greater intra-regional cooperation in providing quality care to their citizens “at all ages”.

### Limitations

While we have attempted comprehensive searches of five academic and one gray literature database, various peak organizations, and sought additional information through personal correspondence, this scoping review was limited to only three out of forty-eight countries in the APAC region that fulfilled the inclusion criteria [9], which are predominately British Commonwealth English-speaking countries.

## 5. Conclusions

Australia, New Zealand, and Singapore use a means-test user charge approach to determine the subsidy provided to RACF residents. The current assessments may provide greater meaningful engagement if more holistic care needs such as spirituality, recreational needs, self-esteem, and autonomy-based needs can be emphasized. The adoption of a holistic approach encompassing a focus on the resident’s spiritual, psychosocial, emotional, cultural, social, and physical needs may assist in promoting the dignity, autonomy, self-esteem, well-being, and quality of life that the people residing in RACFs experience. 

## Figures and Tables

**Figure 1 ijerph-20-07007-f001:**
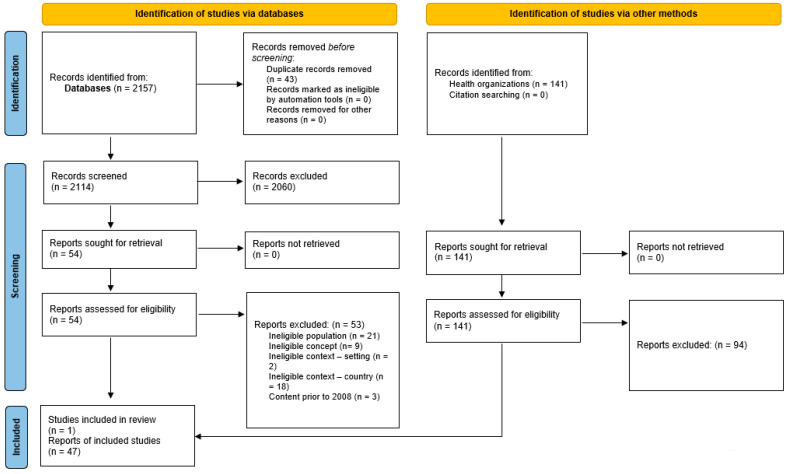
PRISMA-ScR flow diagram of search conducted.

## Data Availability

Data sharing is not applicable as no datasets were generated during the current study.

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
