# Peer review of "Public Mixed Funding for Residential Aged Care Facilities Residents’ Needs in the Asia–Pacific Region: A Scoping Review"

_ijerph, 2023, doi:10.3390/ijerph20217007_

Round 1

Reviewer 1 Report

Comments and Suggestions for Authors

It is better if the abstract is a little more detailed, especially to give an idea of the method and results.

1. What is the main question addressed by the research?In reality, this matter has not been brought up with much importance.
2. Do you consider the topic original or relevant in the field? Does it address a specific gap in the field?
Finding originality in research content makes it difficult to determine research gaps from the literature review.
3. What does it add to the subject area compared with other published material?
The comparative discussion that has been done is very marginal, so a little more elaboration is necessary.
4. What specific improvements should the authors consider regarding the methodology? What further controls should be considered?
The most important issue about the methodology is that no specific process is informed about its analysis.
5. Are the conclusions consistent with the evidence and arguments presented and do they address the main question posed?
Since the main research question is not clear at the outset, it is important to develop the conclusion reasonably.
6. Are the references appropriate?
References that have already been included are linked.
7. Please include any additional comments on the tables and figures.
Considering the above points, it is expected that the article will be improved a lot.

Author Response

The comparative discussion that has been done is very marginal, so a little more elaboration is necessary

Thank you for the constructive feedback. We have also incorporated the other reviewers’ comments and restructured the Discussion section (using track changes) accordingly.

The most important issue about the methodology is that no specific process is informed about its analysis

Thank you for the feedback. In the initial manuscript, we have attempted to reduce word count and refer to our published protocol. The JBI method was employed for this review. The data analysis is fundamentally descriptive and is congruent with the JBI scoping review approach.

Since the main research question is not clear at the outset, it is important to develop the conclusion reasonably.

Thank you for the constructive feedback. Congruent with Reviewer 3 comments for improvement, we have now amended (using track changes) the title, abstract and last paragraph of the Introduction for consistency.

Reviewer 2 Report

Comments and Suggestions for Authors

Thank you for the opportunity to review this paper which, as the authors, state is "forward-looking and aims to inform future discourse in contemplating mixed funding in for RACFs in the APAC region". I believe that the authors have achieved this aim. 

Nevertheless there are a few areas where the paper would benefit from some minor amendment and elaboration in places.

P2 L56. Please avoid hyperbole. Increases in numbers are, by and large, not unprecedented. Labeling them as unprecedented provides a let-off for policy makers who can use this excuse to explain their lack of planning and preparedness.

P3 Figure 1. For consistency (with reports) please provide reasons for why the 94 studies via other methods were excluded.

The discussion would benefit from more depth of explanation in places. For example P7 L294 - 296 states that "The Commission commented that the current system disproportionately impacts people with medium level resources compared with people who have higher-level of assets". This would benefit from a sentence as to WHY the Commission reached this conclusion. 

It is somewhat surprising that both the conclusion and the abstract lay such a strong emphasis on holistic care needs such as spirituality. While this is mentioned in the discussion, the case is not really well made to justify such prominence. If the authors believe that this is justified they should provide a stronger basis in the discussion.

Comments on the Quality of English Language

P1 L39-45. Sentence is too long and becomes incomprehensible. I suggest that the beginning should read "While many countries hypothecate, ..." rather than "With ...".

Author Response

P2 L56. Please avoid hyperbole. Increases in numbers are, by and large, not unprecedented. Labeling them as unprecedented provides a let-off for policy makers who can use this excuse to explain their lack of planning and preparedness.

Thank you for the constructive feedback. We have now amended (using track changes) the text to population aging.

P3 Figure 1. For consistency (with reports) please provide reasons for why the 94 studies via other methods were excluded.

Thank you for the opportunity to clarify the PRISMA-ScR figure. We have amended the text in section 3.1. to be consistent with the Journal and JBI reporting. The 94 reports were excluded as they did not fulfil the inclusion criteria.

The discussion would benefit from more depth of explanation in places. For example P7 L294 - 296 states that "The Commission commented that the current system disproportionately impacts people with medium level resources compared with people who have higher-level of assets". This would benefit from a sentence as to WHY the Commission reached this conclusion.

We have now included more information about the Australian Royal Commission into Aged Care, lines 295-9.

It is somewhat surprising that both the conclusion and the abstract lay such a strong emphasis on holistic care needs such as spirituality. While this is mentioned in the discussion, the case is not really well made to justify such prominence. If the authors believe that this is justified they should provide a stronger basis in the discussion.

Thank you for the constructive feedback, we have now restructured the Discussion section (using track changes) to highlight the importance of funding for non-physical/ non-medical care needs.

P1 L39-45. Sentence is too long and becomes incomprehensible. I suggest that the beginning should read "While many countries hypothecate, ..." rather than "With ...".

Thank you for the constructive comment. We have now redrafted this long sentence accordingly.

Reviewer 3 Report

Comments and Suggestions for Authors

Respected Authors

I have read your article with enthusiasm. The topic chosen by you is in my field of interest. Considering the aging process of the population in the countries of this region, it is very important to review the texts related to aging and related issues. Your article is somewhat well written, but it needs some changes, which are described below.

- Abstract, please add eligibility criteria, date of search, name of databases, and data charting methods.

- Abstract, your aim of the study is not clear enough to understand. Please match your aim with your title and include all elements.

- Abstract, at the beginning of the results, please add data related to the search including total records and number of the included studies. 

- Abstract, please add conclusions that relate to the review questions and objectives.

- Lines 53-4 needs references.

- Please refine your aim at the end of the introduction section. this aim is different from the aim stated in the abstract. Also, both aims do not match with title of the study. 

- Methods, you stated that this review is based on a prior protocol. I have looked at the protocol and there are many differences between it and your study. Is this part of the protocol? Public funding is one of the funding systems. Please clear it.

- Please restructure your methods and results considering the PRISMA Sc-R checklist. Subheading includes; - Protocol and registration, Eligibility criteria, etc.

- Lines 117-18 are related to methods, not results.

- Line 119, based on your statement, you included 48 papers. In my opinion, this statement is not correct. There are several differences between papers, articles, reports, web pages, etc. Please clear the type in the included records. Also, revised Fig 1 based on these clarifications.

- Results, where are the results for "Characteristics of sources of evidence"? Please also restructure your results section considering items of the PRISMA ScR checklist.

Cheers

Comments on the Quality of English Language

There are several punctuation and grammatical errors in the text. 

Author Response

Abstract, please add eligibility criteria, date of search, name of databases, and data charting methods.

Thank you for the constructive feedback. We have appended the PRISMA-ScR fillable checklist to the revised manuscript. We have amended the Abstract accordingly. We have also modelled our Abstract on published examples in the Journal for consistency.

Abstract, your aim of the study is not clear enough to understand. Please match your aim with your title and include all elements.

Thank you for the feedback, we have now amended the Abstract (using track changes) as well as the title to align better with the published protocol and the similar review conducted of European Union countries (Milbank Quarterly 2010:88,286-309).

Abstract, at the beginning of the results, please add data related to the search including total records and number of the included studies.

We have now included the number of included records in the revised Abstract.

Abstract, please add conclusions that relate to the review questions and objectives.

We have constructed the initial Abstract based on the Journal requirements for a single paragraph without headings, a maximum of 200 words. We have now revised the Abstract conclusion to be consistent with the title and review objectives.

Lines 53-4 needs references.

Thank you for the feedback. We have now included the reference, a similar review to ours but focused on European Union countries.

Please refine your aim at the end of the introduction section. this aim is different from the aim stated in the abstract. Also, both aims do not match with title of the study

Thank you for the feedback. We have now amended the title and abstract accordingly to be consistent with the protocol. We have also amended the last paragraph of the Introduction as follows:

The focus on means testing in this review stemmed from the basic principle of social policy on the just allocation of scarce resources based on determinants of health, and the demands on responsible governments to balance competing fiscal demands. [move up from below] Thus, by describing the current funding for residential aged care facility residents’ needs in Asia Pacific countries that provide public mixed funding, this review is forward-looking and aims to inform further discourse on future funding models for RACFs in the APAC region. [move down].

Methods, you stated that this review is based on a prior protocol. I have looked at the protocol and there are many differences between it and your study. Is this part of the protocol? Public funding is one of the funding systems. Please clear it.

Thank you for the constructive feedback, we have now amended line 93 to be more explicit that we are looking for public funding and mean-testing, to be consistent with the protocol.

Please restructure your methods and results considering the PRISMA Sc-R checklist. Subheading includes; - Protocol and registration, Eligibility criteria, etc.

Thank you for the feedback and the affirmation of the importance of sound scientific reporting. We have modelled the manuscript on different scoping reviews published by the Journal. We have also referenced our published protocol. We have now included the PRISM-ScR checklist to ensure that the manuscript is in compliance with PRISMA reporting.

Lines 117-18 are related to methods, not results.

We have modelled the presentation of the search results (lines117-8) based on published examples in the Journal. We have also responded to Reviewer 2’s comments to provide more details about the searches.

Line 119, based on your statement, you included 48 papers. In my opinion, this statement is not correct. There are several differences between papers, articles, reports, web pages, etc. Please clear the type in the included records. Also, revised Fig 1 based on these clarifications.

Thank you for the constructive feedback. We have now amended the text to identify it as “one paper and 47 reports” to be consistent with the PRISMA-ScR figure 1.

Results, where are the results for "Characteristics of sources of evidence"? Please also restructure your results section considering items of the PRISMA ScR checklist.

Thank you for the feedback, we have now appended the PRISMA-ScR checklist to the manuscript.

Round 2

Reviewer 3 Report

Comments and Suggestions for Authors

Respected Authors

Thank you for your modification. In my opinion, your manuscript is acceptable in this fashion.

Thanks